# Longitudinal assessment of changes in hair cortisol levels and associations with violence, poor mental health and harmful substance use among female sex workers in Nairobi, Kenya

**Mamtuti Panneh**[1]*, **Tara Beattie**[2], **Qingming Ding**[3], **Rhoda Kabuti**[4],
**The Maisha Fiti study champions**[4¶], **Polly Ngurukiri**[4], **Mary Kungu**[4],
**Tanya Abramsky**[2], **James Pollock**[5], **Alicja Beksinska**[2], **Erastus Irungu**[4], **Janet Seeley**[2],
**Helen A. Weiss**[6], **Abdelbaset A. Elzagallaai**[3], **Michael J. Rieder**[3], **Rupert Kaul**[7],
**Joshua Kimani**[4], **Mitzy Gafos**[2☯], **John Bradley**[6☯]

1 Department of Infectious Disease Epidemiology and International Health, London School of Hygiene & Tropical Medicine, London, United Kingdom, 2 Department of Global Health and Development, London School of Hygiene & Tropical Medicine, London, United Kingdom, 3 Robarts Research Institute, Schulich School of Medicine and Dentistry, Western University, London, Ontario, Canada, 4 Partners for Health and Development in Africa, Nairobi, Kenya, 5 Department of Immunology, University of Toronto, Toronto, Canada, 6 MRC International Statistics and Epidemiology Group, Department of Infectious Disease Epidemiology, LSHTM, London, United Kingdom, 7 Department of Medicine, University of Toronto, Toronto, Canada

¶ Membership of The Maisha Fiti study champions is listed in the Acknowledgments.
☯ These authors contributed equally to this work.
* Mamtuti.panneh@lshtm.ac.uk, tutipanneh@gmail.com

## Abstract

Female sex workers (FSWs) in sub-Saharan Africa commonly experience violence, mental health problems, and harmful substance use. Stressful life events can harm the functioning of the hypothalamic-pituitary-adrenal (HPA) axis, serving as a pathway to increased poor health, including HIV susceptibility through cortisol levels. In this paper, we examine changes in hair cortisol concentration (HCC) levels and associations with experiences of violence, mental health problems and harmful substance use among FSWs in Nairobi, Kenya. We used baseline and endline data from the Maisha Fiti study of FSWs in Nairobi. Participants reported recent violence, poor mental health, and harmful alcohol/substance use at both time points. Hair samples proximal to the scalp were collected to measure HCC levels determined by ELISA technique. We analysed data from 285 HIV-negative respondents who provided a 2 cm hair sample at baseline and endline. Multivariable linear regression models were used to assess the associations between the trajectory of the main exposure variables and the change in HCC levels at endline. Findings showed that HCC levels decreased significantly (p-value = 0.001) from baseline (mean HCC = 316 ng/g) to endline (mean HCC = 238.1 ng/g). Reported prevalence of violence, mental health problems and harmful alcohol/other substances decreased. There was evidence of

**Data availability statement:** The data are not publicly available due to the need to protect participant confidentiality and safety. Our commitment to participant confidentiality and safety is detailed in our participant information sheet and consent forms, as approved by three ethics committees: Kenyatta National Hospital (P778/11/2018), The London School of Hygiene and Tropical Medicine (16229) and the University of Toronto (37046). Requests to access data should be directed to: research-datamanagement@lshtm.ac.uk, citing item 3643 - The Maisha Fiti Study.

**Funding:** The Maisha Fiti Study is funded by the Medical Research Council MRC and the UK Department of International Development (DFID) (MR/R023182/1) under the MRC/DFID Concordat agreement. MP's PhD is funded by the Commonwealth Scholarship CommisSion in the UK (Award number: GMCS-2020-720). HAW is supported by the MRC and the DFID under the MRC/DFID Concordat agreement and is also part of the EDCTP2 programme supported by the European Union. RK is supported by the Canadian Institute of Health Research (CIHR) grant #PJT-180629 and #PJT-156123. The funders had no role in study design, data collection and analysis, decision to publish, or preparation of the manuscript.

**Competing interests:** The authors have declared that no competing interests exist.

associations between change in HCC at endline and the trajectories of physical violence (p-value = 0.007) and physical and/or sexual violence (p-value = 0.048). There was weak evidence of an association between the trajectory of exposure to emotional violence but no evidence of other associations. These findings suggest that physical violence and physical and/or sexual violence may lead to HPA axis dysfunction, possibly serving as a pathway linking violence to increased poor health, including HIV acquisition. However, further research with repeated measurements and a larger sample size is needed to examine the associations between violence, HCC levels, and HIV infection.

## Introduction

Female sex workers (FSWs) in sub-Saharan Africa (SSA) are at heightened risk of adverse life events such as violence, poor mental health and harmful alcohol and other substance use, which have been associated with negative health outcomes, including increased risk of HIV acquisition [1,2]. Vulnerabilities of FSWs to violence, poor mental health and harmful alcohol and other substance use have been linked to the criminalisation of sex work, the risks inherent in the sex work environment, as well as the intersecting socio-economic and structural inequalities (e.g., poverty, low education and gender inequality) they face from as early as childhood [2–4]. According to the UNAIDS, the risk of HIV was 26 times higher among FSWs worldwide in 2020 compared to women of the same age not engaged in sex work [5].

Similar to other SSA countries, FSWs in Kenya are at an increased risk of adverse life experiences, mainly linked to poverty, stigma and discrimination, gender inequality, the criminalisation of sex work and high-risk working conditions [1,6,7]. Although violence against women is generally high in Kenya, with 47% of women aged 15–49 years ever experiencing physical and/or sexual violence in their lifetime, FSWs tend to report significantly higher rates. For example, a recent finding from a cross-sectional study conducted among FSWs in Nairobi found that approximately 90% experienced violence in the past 12 months. Of these, 81% reported violence from clients, 79% from intimate partners, and 50% from other perpetrators (e.g., police, family members)[8]. Baseline findings from the Maisha Fiti study of FSWs in Nairobi, from which participants for this study were drawn, indicated that nearly 50% reported symptoms of depression, 38% reported symptoms of anxiety, and 30% engaged in harmful alcohol consumption [4].

There is strong evidence of associations between the risk of HIV infection and violence [9,10] mental health problems [11] and harmful alcohol and other substance use [12–14]. Although violence, poor mental health and harmful alcohol and/or other substance use can lead to higher-risk sexual behaviours (e.g., condomless sex), increasing FSWs' vulnerability to HIV infection, there may be other possible pathways to increased HIV acquisition [11,15,16]. For example, a cross-sectional study among HIV-negative women showed that physical and psychological intimate partner violence independent of sexual abuse was associated with immune activation

(CD4 + activation), suggesting an immune link between violence and increased HIV susceptibility [17]. Also, previous research indicates that the experience of traumatic or stressful events may increase the susceptibility to HIV infection through biological or physiological pathways [18,19]; however, these potential pathways are less understood.

Recent empirical studies suggest that stressful life events can harm the functioning of the hypothalamic-pituitary-adrenal (HPA) axis, which may serve as a pathway linking stressful life experiences and negative health outcomes [20]. Activation of the HPA axis due to physiological (e.g., illness, injury, or trauma) or psychological (e.g., mental-ill health) stress leads to the production of the stress hormone, cortisol [21]. During acute stress, cortisol is released for several hours, and once the optimal cortisol concentration is achieved, it exerts negative feedback and returns to systemic homeostasis [21]. However, long-term or repeated stress results in a dysregulated HPA axis, often characterised by hyper-or-hypo-responsiveness, which disturbs cortisol production and might lead to a condition called allostatic load, the 'wear and tear' on the body [22,23]. Research has shown that hypercortisolism mostly occurs in the early onset of trauma or stress and then lapses to reduced levels (hypocortisolism) over time as a "maladaptive response" to protect the body [22,24,25]. Dysregulation in cortisol production due to stress is known to have a deleterious impact on health, as most bodily cells contain cortisol receptors [21]. For example, cortisol plays a key role in immune regulation, including the anti-inflammation processes, and hypocortisolism might lead to increased levels of inflammatory cytokines, causing widespread inflammation [26]. Also, although cortisol acts as an anti-inflammatory hormone in normal homeostasis, hyper-cortisolism can have proinflammatory effects due to glucocorticoid receptor resistance, which has been linked to failure to downregulate inflammatory responses to viruses [26,27]. Dysregulation in cortisol levels has been associated with higher T-cell activation (e.g., CD8 + T-cells), and research has shown that individuals with activated immune markers are more susceptible to HIV infection [28,29].

A growing body of evidence from different populations suggests associations between violence, mental health problems and harmful alcohol and other substance use with cortisol levels, although results have been inconsistent [30–33]. Among FSWs, a recent cross-sectional study in Kenya showed that FSWs who recently (in the past 12 months) experienced gender-based violence had higher hair cortisol levels than their unexposed counterparts, but no evidence of association was found for depression, post-traumatic stress disorder (PTSD) or harmful alcohol and other substance use [19]. Also, baseline findings of FSWs in this current study cohort showed evidence of associations of both physical and/or sexual violence and harmful alcohol and/or other substance use with increased hair cortisol levels [34]. Currently, there is a huge gap in the longitudinal understanding of cortisol levels in relation to violence, poor mental health and harmful alcohol and other substance use among FSWs. In several studies, cortisol is measured using blood, saliva, or urine, which only measure short-term cortisol levels. This makes it hard to assess the relationship between long-term stress exposure and the functioning of the HPA axis [35]. In contrast, analysing cortisol in hair provides reliable long-term cortisol measurement retrospectively for several months [35,36].

In this study, we used data from the Maisha Fiti study, a mixed-methods longitudinal study with FSWs in Nairobi. The Maisha Fiti study aimed to investigate the associations of violence, poor mental health, harmful alcohol and other substance use and the biological changes to the immune system and risk of HIV [4]. Behavioural-biological surveys were conducted in the Maisha Fiti study, with baseline data collected before the start of the COVID-19 pandemic and endline data collected after the COVID-19 pandemic lockdown in Kenya. In this paper, we use the baseline and endline data to examine whether hair cortisol levels change over time from baseline to endline and if hair cortisol levels changed based on the trajectory of exposure to different types of violence, mental health problems and harmful alcohol and other substance use among HIV-negative FSWs in Nairobi from baseline to endline. To our knowledge, this study will be the first to examine cortisol levels among FSWs longitudinally. The overarching goal is to understand whether the experiences of violence, poor mental health, and harmful use of alcohol and other substances affect the stress response system through cortisol levels, which could be a possible pathway linking these exposures to increased risks of poor health, including HIV infection.

## Methods

### Study design and sampling

The Maisha Fiti longitudinal study was designed in consultation with the community of FSWs in Nairobi and the staff and peer educators working in the seven Sex Work Outreach Program (SWOP) clinics in Nairobi. Baseline data were collected from 14 June to 13 December 2019 before the onset of the COVID-19 pandemic, while endline data were collected between 11 June 2020 and 29 January 2021 after the COVID-19 lockdown in Kenya.

The SWOP clinics in Nairobi provide clinical services to 73% (29,000) of the estimated 39,600 FSWs in Nairobi. The participants for the Maisha Fiti study were randomly selected across the seven SWOP clinics in Nairobi using their unique SWOP clinic enrolment numbers. The eligibility criteria for the selection of study participants in the Maisha Fiti study were as follows:

(i) aged 18–45 years, (ii) attended a SWOP clinic in the past 12 months (as an indication of current sex work), and (iii) no chronic illness (excluding HIV) such as diabetes, rheumatoid arthritis, asthma, and TB infection that could impact the immune system. A total of 10,292 of the 29,000 FSWs met these criteria and were included in the sampling frame. The desired sample size for the Maisha Fiti study was 1000 FSWs; however, 1200 FSWs were randomly selected from the 10,292 FSWs (who met the inclusion criteria) to allow for non-response and non-eligibility since other additional exclusion criteria were assessed during enrolment including current pregnancy (urine samples were tested for pregnancy) or breastfeeding. Also, the number of FSWs selected per clinic was proportional to clinic size, and women aged less than 25 years were oversampled to allow adequate power for analyses stratified by age. Selected participants were telephoned and informed about the study. Those interested were invited to visit the study clinic, where they received detailed information about the study. Participation was voluntary; eligible participants provided written informed consent before completing the behavioural-biological survey. Details about the study sampling have been published elsewhere [37]. This current analysis focused on the HIV-uninfected participants in the Maisha Fiti study.

### Data collection

**Behavioural-biological surveys.** Participants completed a behavioural-biological survey at each study visit. The behavioural survey captured data on socio-demographics, adverse childhood experiences (ACEs), sexual practices and behaviours, stigma, social support, lifetime and recent violence experiences, mental health problems, and alcohol and other substance use. However, some time-invariant sociodemographic factors (e.g., age, marital status, number of children, religion, literacy, socioeconomic status) were only collected at baseline. Biological samples, including urine, blood, and hair samples, were collected for biological tests. Urine samples were provided to test for gonorrhoea and chlamydia infection using GeneXpert Assay. Blood samples were used to test for HIV and syphilis using rapid HIV tests and rapid plasma regain assay, respectively, and positive HIV results were confirmed using HIV DNA GeneXpert. Hair samples were used to test for cortisol levels. Details about the processing of hair cortisol can be found in the study variable section below.

### Ethical considerations

The Maisha Fiti study was ethically approved by the Research Ethics Committees at the London School of Hygiene and Tropical Medicine (Approval number: 16229), the Kenyatta National Hospital – University of Nairobi Ethics Review Committee (KNH ERC P778/11/2018), and the University of Toronto (Approval number: 37046). Participation in the study was voluntary, and women provided written informed consent before enrolling. Confidentiality was maintained throughout the study, and anonymised study numbers were issued to each participant.

### Conceptual framework

Using the eco-social life course theory [38] and drawing on the literature, a conceptual framework was developed to guide our analyses (Fig 1). The framework theorises how societal and ecological context exposures can be biologically

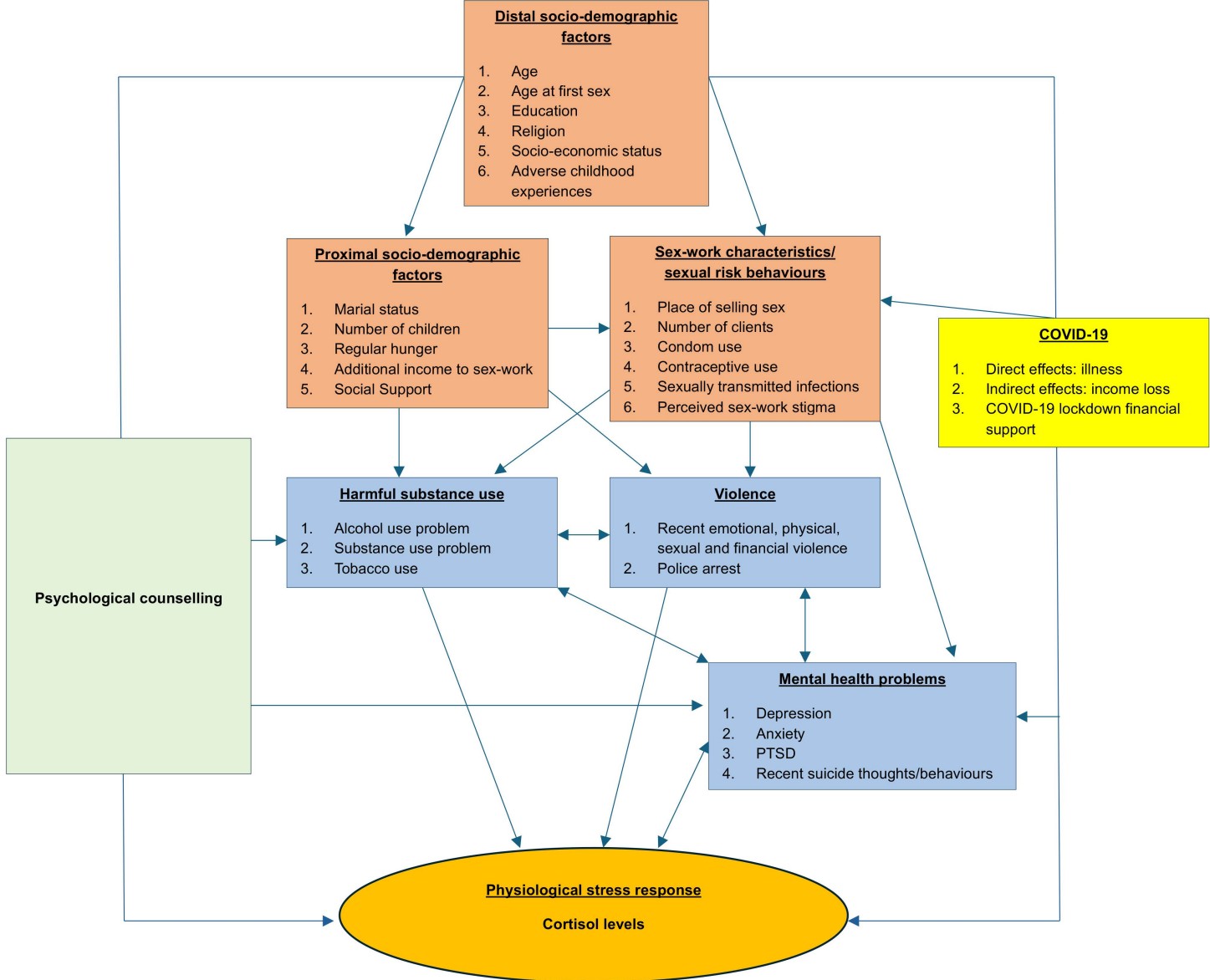

**Fig 1. A conceptual framework illustrating the risk factors influencing hair cortisol levels among female sex workers in Nairobi, Kenya, over time.**

embodied, causing health and disease disparities [38]. It is based on our baseline analysis and has been adapted to include the impact of attending psychological counselling from the Maisha Fiti study counsellor, SWOP clinic, or any healthcare provider prior to endline and the effect of COVID-19 on cortisol levels over time. The direct effect of COVID-19 through illness was hard to determine since most participants with flu-like symptoms weren't tested for COVID-19 at that time, as there was a shortage of COVID-19 test kits in Kenya. Also, the indirect effect of COVID-19 through income loss was not directly measured, but the trajectory of recent hunger could be in proximity to it. The rectangles in blue are the main exposure variables of interest, while the other rectangles are potential confounders, with the effect of counselling being a possible effect modifier. Details about the definition of some variables in the framework are shown in Table 1.

**Table 1. Definition of exposure variables.**

| Variables | Tool/Question | Category |
|---|---|---|
| Childhood and distal Socio-demographic factors | | |
| ACEs | WHO Adverse Childhood Experiences International Question-naire (ACE-IQ) [39] | Ordered categorical variable (each ACE scores one point): < 4, 5–8, 9–12 [4] |
| Age at first sex | How old were you when you first received money/goods in exchange for sex? | </= 15, 16-17, 18+ |
| Socio-economic status (SES) | 14 household asset questions used in the Kenyan Demographic Health Surveys | Principle component analysis (PCA) used to compute household SES: Lower/lower middle, middle, upper middle/upper. |
| Proximal Socio-demographic Factors | | |
| Number of household dependants | Not including yourself, how many people living in your house-hold are dependent on your income? | 0, 1, 2+ |
| Recent hunger | Thinking now about the past 7 days, have you or anyone in your family skipped a meal because there was not enough food? | No vs Yes |
| Current social support | Do you have someone who you can talk to about your problems? | Yes/sometimes vs No |
| Sex-work characteristics/sexual risk behaviours | | |
| Condom use | The last time you had vaginal sex, did you use a condom? | No vs Yes |
| Contraceptive use | Are you currently doing something or using any method to delay or avoid getting pregnant? What is the method you currently use to delay or avoid getting pregnant? | No vs Yes |
| Recent Violence (past six months) | | |
| Recent physical violence, sexual violence and emotional Violence | WHO Violence Against Women 13-item questionnaire [40] | No vs Yes for each type of violence |
| Financial Violence | In the last six months, how many times has a client refused or had to be forced to pay you for sex you have provided? | Never vs ≥ once |
| Recent Police arrest | Have you been arrested in the past six months because you are a sex worker? | No vs Yes |
| Mental health Problems | | |
| Depression (past 2 weeks) | Patient Health Questionnaire-9 (PHQ-9) [41,42] | < 15 = none/mild vs ≥ 15 = moderate/severe |
| Anxiety (past 2 weeks) | Generalised Anxiety Disorder-7 Assessment (GAD-7) [41,43] | < 10 = none/mild vs ≥ 10 = moderate/severe |
| PTSD (past month) | Harvard Trauma Questionnaire (HTQ-17) [44] | < 2 = negative PTSD vs ≥ 2 = positive for PTSD |
| Recent Suicidal Thoughts/Behaviours (past month) | Having thoughts about ending your life Having attempted to end your life | No vs Yes |
| Alcohol and other substance use problems (past 3 months) | | |
| Harmful alcohol | WHO ASSIST (Alcohol, Smoking and Substance Involvement Screening Test) tool [45] | score ≥ 11 moderate to high risk |
| Other harmful substance | WHO ASSIST (Alcohol, Smoking and Substance Involvement Screening Test) tool [45] | score ≥ 4 moderate to high risk |
| Tobacco use | In your life have you ever used Tobacco products? In the past 3 months, how often have you used tobacco? | Never vs once/twice/monthly/weekly/daily or almost every day |
| | **Psychological counselling/COVID-19** | |
| Psychological counselling | In the past 6 months, have you had one or more counselling sessions with the Maisha Fiti study counsellor? In the past 6 months, did you receive psychological counselling from SWOP clinic? In the past 6 months, did you receive psychological counselling from any other service provider? | No vs Yes |
| COVID-19 lockdown financial support | Since the COVID-19 lockdown, did you receive any funds, food, or vouchers from a government scheme? | No vs Yes |

## Study variables

**Outcome variable.** The main outcome variable was hair cortisol concentration (HCC), with change in HCC at endline generated by dividing endline HCC by baseline HCC levels. This means that if the change in HCC at endline has a value greater than 1, it indicates an increase in HCC at endline compared to baseline. Conversely, a value less than 1 indicates a decrease in HCC at endline compared to baseline, while a value equal to 1 means there was no change, i.e., HCC was the same at both baseline and endline.

For each participant who agreed to provide a hair sample, about 50 hair strands were cut from the posterior vertex, next to the scalp, using clean scissors. The scalp end of the hair sample was labelled, packaged in aluminium foil, and stored in a cool, dry place. Of the 1003 women recruited to participate in the Maisha Fiti study, 746 were HIV-negative at baseline [4], of whom 736 (98.7%) provided hair samples. However, 425 women at baseline had a useable sample; of these, 285 had useable follow-up samples at endline. This is because several participants had extremely short hair or hair samples with few hair strands. This led to the exclusion of hair samples less than 2 cm in length (a hair length of about 6 cm was ideal) to allow for sufficient samples to proceed with hair cortisol testing assays. The proximal 2cm of hair from the scalp was analysed for cortisol concentration and represents cumulative cortisol concentration over 2.5 months based on the average African hair growth rate of 0.79 cm per month [19]. Hair cortisol was analysed using an established enzyme-linked immunosorbent (ELISA) technique [46] and expressed as nanogram/gram (ng/g) of hair mass.

**Exposure variables.** The time-variant measures that were collected at the two time points were recent violence of different forms, mental health problems, including suicidal behaviours, alcohol and other substance use problems, sex work characteristics, sexual risk behaviours and recent hunger as an indication of financial stress. The trajectories of all time-variant exposures were categorised into three levels: never (for both baseline and endline), baseline only, and endline (with or without baseline). The time-invariant variables collected at baseline only were the distal socio-demographic factors and proximal socio-demographic factors (except recent hunger), while the experience of psychological counselling and the impact of COVID-19 were collected at endline.

The three categories of our main exposure variables of interest were the trajectories of recent exposures to (i) violence of different forms, (ii) mental health problems, and (iii) harmful alcohol and other substance use. The category of recent violence includes the different forms of violence, such as physical, sexual, emotional, and financial violence, and the trajectories of each of these violence variables were created. We further created a combined variable for experiencing physical and/or sexual violence because this combination is often used in other cohorts of FSWs and because research showed that women who experience physical and/or sexual violence are more likely to have multiple health problems, including increased HIV infection [47]. The trajectory of mental health problems involved participants who experienced any mental health problems (depression/anxiety/PTSD/suicidal behaviour). The category of alcohol and other substance use includes the trajectories of harmful alcohol use, other harmful substance use, and tobacco use, all treated as separate variables. Tobacco use was not included in the list of other harmful substances because it was assessed differently, as shown in Table 1.

In the conceptual framework above in Fig 1, all the variables upstream of the main exposures and outcome variables were considered covariates.

## Statistical analyses

Descriptive statistics were conducted for each exposure variable, and frequencies and percentages were presented. Due to the skewed distribution of HCC levels for both time points and skewness in the change in HCC at endline, they were log-transformed for all statistical analyses. We examined the mean HCC for baseline and endline and determined if there was a significant mean difference in HCC levels across these time points using a paired t-test. The mean of change in HCC at endline with respect to each exposure variable was also reported. For easy interpretation, the mean of HCC levels for both baseline and endline, as well as the mean of change in HCC at endline in descriptive analyses, were exponentiated to produce a geometric mean, the backwards-transformed mean from the log-transformed data.

To examine whether cortisol levels changed between baseline and endline based on the experiences of our main exposure variables (recent violence, poor mental health, and alcohol or substance use), initial associations between change in HCC at endline and the trajectories of our main exposures, as well as covariates, were assessed using simple linear regression. Key exposures that were associated with change in HCC at endline in univariate linear regression analyses (p<0.1) were taken forward for multivariable linear regression analyses and adjusted for confounders that were associated with a change in endline HCC (p<0.1) in univariate analyses. To avoid multicollinearity, we examined in separate multivariable regression models how the trajectories of the different forms of violence, mental health problems, and alcohol and other substance use were associated with a change in endline HCC, adjusting for confounders with significance set at p<0.05. In each multivariable regression model, we assessed for effect modification with the uptake of counselling by testing if the uptake of counselling prior to endline modified the relationships between the key exposures and change in HCC at endline. The coefficients from linear regression models were exponentiated to produce geometric mean ratios (GMR), the ratio of the means that have been backward transformed. All analyses were performed using STATA 16.1, weighted for age and adjusted for clustering by clinic, with p-values obtained using the adjusted Wald test. Additionally, variables with > 5% missing observations were reported.

## Results

Among the 285 participants with follow-up cortisol measurements, the mean age was 31.9 years (CI: 31.3-32.6), with most participants in the age bracket 25–34 years (42.2%). The mean age of sexual debut was 16.6 years (CI: 16.2-16.9), with 30.5% reporting first sex at 15 years or younger. Most participants were literate (87.1%) and Christians (Protestants 52.8%; Catholics 40.8%). The majority (66.3%) had been divorced, widowed, or separated from their partners, and almost all (95.2%) had at least one child. Most participants (80.7%) had one or more household members depending on their income, and more than two-thirds (69.4%) reported having someone to talk to about their problems. Almost half had received counselling from either the Maisha Fiti study counsellor, SWOP clinic or other healthcare providers, and about 18.9% received COVID-19 support from the government (Table 2). The characteristics of these study participants are similar to the 425 HIV-negative participants with cortisol results at baseline as well as the overall baseline study participants of the Maisha Fiti study published elsewhere [4].

Compared to baseline, the overall prevalence of all the different forms of violence, mental health problems and harmful alcohol and other substance use decreased at endline (Table 3). For time-varying co-variates, there was an increase in prevalence from baseline to endline for some, such as recent hunger (28.1% vs 51.7%), while a decrease was seen in others, such as client volume per week (>5 clients/week: 40.3% vs 17.5%) and sex-work-related stigma (86.78% vs 61.47%). More details about the change in the prevalence of time-varying variables can be found in Table 3. The trajectory changes in the experiences of the different forms of violence, mental health problems and harmful alcohol and other substances from baseline to endline are also presented in Table 3. For example, 26.9% of participants did not experience financial violence across the two time points, 37.7% experienced financial violence at baseline only, and 35.4% experienced financial violence at endline (with/without baseline financial violence). For emotional violence, 16.2% did not experience emotional violence, 40.0% at baseline only, and 43.8% at endline.

The mean HCC at baseline and endline were 316.0 ng/g (95% CI: 271.9-367.3) and 238.1 ng/g (95% CI: 211.2-268.4), respectively, with evidence of a significantly lower HCC at endline compared to baseline (p=0.001). For the trajectories of our main exposure variables, shown in Table 3, the trajectories of exposure to recent emotional (p-value=0.012), physical (p-value=<0.001), and physical and/or sexual violence (p-value=<0.009) showed evidence of associations with change in HCC at endline in univariate analyses, while weak evidence was found for alcohol use problems (p-value=0.074). The time-varying covariate that showed evidence of an association with change in HCC in univariate regression was condom use at last sex (p-value=0.013), with women who reported condom use at endline having higher HCC levels at endline (crude GMR 1.79; 95% CI: 1.21 to 2.64) compared to those who reported condom none-use at last sex throughout the

**Table 2. Socio-demographic characteristics of study participants (n = 285).**

| Study variables | N (%) | Geometric mean of the change in HCC at endline[a] | Crude geometric mean ratio (95% CI) | P-value* |
|---|---|---|---|---|
| Age | | | | 0.394 |
| <25 | 90 (18.6) | 0.87 | Reference | |
| 25-34 | 101 (42.2) | 0.67 | 0.78 (0.53 to 1.13) | |
| 35+ | 94 (39.3) | 0.80 | 0.92 (0.62 to 1.38) | |
| Age at first sex | | | | 0.872 |
| </= 15 | 88 (30.5) | 0.76 | Reference | |
| 16-17 | 97 (33.0) | 0.79 | 1.05 (0.71 to 1.54) | |
| 18+ | 97 (36.5) | 0.71 | 0.94 (0.63 to 1.40) | |
| Literacy | | | | 0.567 |
| illiterate | 35 (12.9) | 0.84 | Reference | |
| literate | 250 (87.1) | 0.74 | 0.89 (0.59 to 1.34) | |
| Religion | | | | 0.508 |
| Catholic | 118 (40.8) | 0.82 | Reference | |
| Protestant | 144 (51.3) | 0.74 | 0.91 (0.66 to 1.26) | |
| Muslim/others/none | 23 (7.9) | 0.54 | 0.66 (0.32 to 1.36) | |
| Socio-economic status | | | | 0.458 |
| Lower/lower middle | 105 (35.8) | 0.80 | Reference | |
| Middle | 53 (17.9) | 0.61 | 0.77 (0.50 to 1.18) | |
| Upper middle/upper | 127 (46.3) | 0.78 | 0.99 (0.69 to 1.39) | |
| Marital status | | | | 0.352 |
| Single | 91 (27.7) | 0.83 | Reference | |
| Married or cohabiting | 16 (6.1) | 1.09 | 1.30 (0.61 to 2.78) | |
| Separated/divorced/widowed | 178 (66.3) | 0.70 | 0.84 (0.58 to 1.21) | |
| Total number of ACEs reported | | | | 0.058 |
| 0-4 | 79 (27.3) | 1.01 | Reference | |
| 5–8 | 161 (57.1) | 0.70 | 0.70 (0.50 to 0.98) | |
| 9–12 | 45 (15.6) | 0.58 | 0.57 (0.32 to 1.02) | |
| Number of children | | | | 0.0032 |
| 3+ | 60 (25.9) | 1.08 | Reference | |
| 1-2 | 188 (69.3) | 0.63 | 0.58 (0.42 to 0.80) | |
| None | 17 (4.8) | 1.17 | 1.08 (0.43 to 2.73) | |
| Number of household dependents | | | | 0.166 |
| 0 | 62 (19.3) | 0.54 | Reference | |
| 1 | 78 (25.8) | 0.83 | 1.52 (0.95 to 2.44) | |
| 2+ | 145 (54.8) | 0.81 | 1.48 (0.95 to 2.30) | |
| Social support | | | | 0.210 |
| No | 89 (30.6) | 0.66 | Reference | |
| Yes | 196 (69.4) | 0.80 | 1.22 (0.89 to 1.66) | |
| Place of selling sex at baseline | | | | |
| Lodge/hotel/rented room/home | 272 (96.8) | 0.73 | Reference | |
| Public places | 9 (3.2) | 1.13 | 1.53 (0.47 to 5.00) | 0.479 |
| *Collected at endline* | | | | |
| Had counselling | | | | 0.871 |
| No | 152 (52.1) | 0.74 | Reference | |
| Yes | 133 (47.9) | 0.76 | 1.03 (0.75 to 1.41) | |

*(Continued)*

**Table 2.** (Continued)

| Study variables | N (%) | Geometric mean of the change in HCC at endline[a] | Crude geometric mean ratio (95% CI) | P-value* |
|---|---|---|---|---|
| Received COVID-19 financial support | | | | 0.869 |
| No | 228 (81.1) | 0.76 | **Reference** | |
| Yes | 57 (18.9) | 0.73 | 0.97 (0.63 to 1.48) | |

[a] Geometric mean is the backward-transformed mean from the transformed data.

* P-value obtained using simple linear regression.

two-time points. Moreover, the time-invariant variable (Table 2) that showed evidence of associations with change in HCC at endline was number of children (p = 0.0032), with those with one to two children recording significantly lower HCC at endline (crude GMR 0.58; 95% CI: 0.42 to 0.80) compared to those with three or more children. Weak evidence of an association was found between the total number of ACEs reported and change in HCC (p-value = 0.058).

In adjusted analysis (Table 4), we found weak evidence of an association between the trajectory of emotional violence and change in HCC at endline (p-value = 0.085) (**Model 1**), with participants who experienced emotional violence at baseline only having insignificantly lower endline HCC (adjusted GMR 0.69; 95% CI 0.42-1.13) and those that reported emotional violence at endline having significantly lower endline HCC (adjusted GMR 0.58; 95% CI 0.36-0.94) compared to their unexposed counterparts. There was evidence of an association between the trajectories of exposure to recent physical violence (p-value = 0.007) (**Model 2**) and physical and/or sexual violence (p-value = 0.048) (**Model 3**) with change in HCC at endline. Participants who experienced physical violence only at baseline had a significantly decreased HCC at endline (adjusted GMR 0.59; 95% CI 0.42 -0.82) compared to those who did not experience physical violence. Also, compared to participants who didn't report physical and/or sexual violence throughout the two-time points, those who reported physical and/or sexual violence at baseline only (adjusted GMR 0.71; 95% CI 0.50- 1.00) and at endline (adjusted GMR 0.63; 95% CI 0.42-0.96) had significantly lower endline HCC. In addition, no evidence of an association was found between alcohol use problems and change in HCC at endline (p-value = 0.239) in multivariate regression (**Model 4**).

When we examined the interaction of counselling, there was evidence that the association between the trajectories of exposure to physical violence and physical and/or sexual violence, with decreased HCC at endline, were stronger amongst women who attended counselling compared to those who did not attend counselling (Table 4). For example, among counselled women, those who experienced physical violence at baseline only (adjusted GMR 0.39; 95% CI 0.23-0.66) or at endline (adjusted GMR 0.36; 95% CI 0.20-0.64) had significantly lower HCC at endline compared to women who didn't experience counselling. Similar findings were also found among counselled women with trajectories of exposure to physical and/or sexual violence. Among participants not exposed to counselling, HCC levels were lower (adjusted GMR 0.77; 95% CI 0.48 to 1.22) and higher (adjusted GMR 1.16; 95% CI 0.51 to 2.65) in those who experienced physical violence at baseline only and at the endline, respectively, but results were not significant. Similar patterns were found among those who experienced physical and/or sexual violence. In addition, we found weak (p-value = .0097) and no evidence (p-value = 0.142) that counselling modified the effect of the associations between emotional violence and alcohol use problem with a change in HCC at endline, respectively.

## Discussion

In this longitudinal study of FSWs in Nairobi, we found a significant decrease in mean HCC level from baseline to endline. Within the same period, there was a decrease in the prevalence of the different forms of violence (physical, sexual, emotional, and financial violence), mental health problems and harmful alcohol and other substance use. Findings from the multivariable linear regression models showed evidence of associations between the trajectory of exposures to physical

**Table 3.** Change in HCC at endline in relation to change in violence, poor mental health, alcohol, and other substance use problems and other trajectory variables (n=285).

| Variables | N (Weighted %) | Change in Prevalence (BL vs EL) | Geometric mean[a] of the change in HCC at endline | Crude geometric mean ratio (95% CI) | P-value* |
|---|---|---|---|---|---|
| Financial Violence | | 65.99 vs 35.09 | | | 0.253 |
| Never | 81(26.9) | | 0.93 | Reference | |
| Baseline only | 104 (37.7) | | 0.69 | 0.74 (0.50 to 1.09) | |
| Endline (with/without baseline) | 96 (35.4) | | 0.70 | 0.75 (0.40 to 1.13) | |
| Emotional Violence | | 79.19 vs 43.75 | | | 0.012 |
| Never | 50 (16.2) | | 1.23 | Reference | |
| Baseline only | 111 (40.0) | | 0.78 | 0.64 (0.40 to 1.01) | |
| Endline (with/without baseline) | 124 (43.8) | | 0.61 | 0.49 (0.31 to 0.79) | |
| Physical Violence | | 53.77 vs 20.20 | | | |
| Never | 117 (40.0) | | 1.10 | Reference | 0.000 |
| Baseline only | 109 (39.81) | | 0.56 | 0.51 (0.36 to 0.72) | |
| Endline (with/without baseline) | 59 (20.2) | | 0.63 | 0.57 (0.37 to 0.88) | |
| Sexual violence | | 47.10 vs 17.71 | | | 0.689 |
| Never | 135 (46.0) | | 0.81 | Reference | |
| Baseline only | 100 (36.3) | | 0.71 | 0.88 (0.61 to 1.26) | |
| Endline (with/without baseline) | 50 (17.7) | | 0.69 | 0.85 (0.55 to 1.32) | |
| Physical and/ or sexual violence | | 65.02 vs 28.75 | | | 0.009 |
| Never | 87 (28.5) | | 1.08 | Reference | |
| Baseline only | 116 (42.7) | | 0.67 | 0.62 (0.43 to 0.89) | |
| Endline (with/without baseline) | 82 (28.8) | | 0.63 | 0.58 (0.40 to 0.87) | |
| Recent arrest | | 29.59 vs 15.84 | | | 0.1723 |
| Never | 183 (63.5) | | 0.70 | Reference | |
| Baseline only | 59 (20.6) | | 1.01 | 1.44 (0.96 to 2.15) | |
| Endline (with/without baseline) | 43 (15.8) | | 0.67 | 0.95 (0.62 to 1.46) | |
| Any mental Health problem | | 29.59 vs 11.68 | | | 0.72 |
| Never | 189 (64.8) | | 0.79 | Reference | |
| Baseline only | 65 (23.6) | | 0.69 | 0.87 (0.60 to 1.26) | |
| Endline (with/without baseline) | 31 (11.7) | | 0.70 | 0.89 (0.54 to 1.45) | |
| Alcohol use problem | | 35.09 vs 17.08 | | | 0.074 |
| Never | 165 (58.2) | | 0.83 | Reference | |
| Baseline only | 68 (24.6) | | 0.54 | 0.65 (0.44 to 0.95) | |
| Endline (with/without baseline) | 50 (17.2) | | 0.81 | 0.97 (0.64 to 1.46) | |
| Other substances | | 35.6 vs 32.06 | | | 0.929 |
| Never | 152 (44.4) | | 0.73 | Reference | |
| Baseline only | 37 (12.5) | | 0.79 | 1.08 (0.67 to 1.75) | |
| Endline (with/without baseline) | 96 (32.1) | | 0.77 | 1.05 (0.73 to 1.52) | |
| Tobacco | | 21.87 vs 15.42 | | | 0.927 |
| Never | 213 (75.2) | | 0.74 | Reference | |
| Baseline only | 28 (9.4) | | 0.82 | 1.11 (0.52 to 2.36) | |
| Endline (with/without baseline) | 44 (15.4) | | 0.79 | 1.07 (0.69 to 1.66) | |
| Recent hunger | | 28.14 vs 51.68 | | | 0.833 |
| Never | 119 (40.4) | | 0.80 | Reference | |
| Baseline only | 21 (7.9) | | 0.70 | 0.87 (0.50 to 1.52) | |

*(Continued)*

**Table 3.** (Continued)

| Variables | N (Weighted %) | Change in Prevalence (BL vs EL) | Geometric mean[a] of the change in HCC at endline | Crude geometric mean ratio (95% CI) | P-value* |
|---|---|---|---|---|---|
| Endline (with/without baseline) | 145 (51.7) | | 0.73 | 0.91 (0.65 to 1.29) | |
| Have other source (s) of income | | 53.11 vs 52.71 | | | 0.696 |
| Never | 72 (25.6) | | 0.69 | Reference | |
| Baseline only | 59 (21.3) | | 0.72 | 1.04 (0.65 to 1.67) | |
| Endline only | 154 (53.1) | | 0.80 | 1.17 (0.77 to 1.76) | |
| >5 clients/week | | 40.30 vs 17.54 | | | 0.368 |
| Never | 148 (52.9) | | 0.83 | Reference | |
| Baseline only | 83 (29.4) | | 0.66 | 0.80 (0.56 to 1.14) | |
| Endline (with/without baseline) | 49 (17.8) | | 0.66 | 0.79 (0.52 to 1.20) | |
| Condom use last sex | | 26.66 vs 29.16 | | | 0.013 |
| Never | 33 (10.8) | | 0.46 | Reference | |
| Baseline only | 45 (15.8) | | 0.66 | 1.42 (0.90 to 2.23) | |
| Endline (with/without baseline) | 207 (73.3) | | 0.83 | 1.79 (1.21 to 2.64) | |
| Contraceptive use | | 85.83 vs 87.52 | | | 0.325 |
| Never | 16 (5.0) | | 0.51 | Reference | |
| Baseline only | 23 (7.5) | | 0.79 | 1.55 (0.70 to 3.43) | |
| Endline (with/without baseline) | 246 (87.5) | | 0.77 | 1.50 (0.88 to 2.56) | |
| Bacterial STI (Chlamydia/Gonorrhoea/syphilis) | | 13.54 vs 12.72 | | | 0.994 |
| Never | 217 (77.3) | | 0.75 | Reference | |
| Baseline only | 28 (10.0) | | 0.73 | 0.96 (0.46 to 2.00) | |
| Endline (with/without baseline) | 39 (12.7) | | 0.76 | 1.01 (0.64 to 1.58) | |
| Experienced abortion/still birth** | | 45.1 vs 3.12 | | | 0.936 |
| Never | 146 (53.4) | | 0.73 | Reference | |
| Baseline only | 110 (43.4) | | 0.77 | 1.06 (0.75 to 1.48) | |
| Endline (with/without baseline) | 9 (3.3) | | 0.84 | 1.16 (0.28 to 4.79) | |
| Reports any sex-work-related stigma | | 86.78 vs 61.47 | | | 0.419 |
| Never | 23 (8.0) | | 0.55 | Reference | |
| Baseline only | 88 (30.5) | | 0.80 | 1.46 (0.83 to 2.59) | |
| Endline (with/without baseline) | 171 (61.6) | | 0.76 | 1.39 (0.80 to 2.41) | |

[a] Geometric mean is the backward-transformed mean from the transformed data.

* P-value obtained using simple linear regression.

** Missing n = 20.

violence and physical and/or sexual violence with change in HCC at endline. However, findings showed that the uptake of counselling modified these relationships. Lastly, we found no evidence of an association between the trajectories of financial violence, harmful alcohol, and other substances with change in HCC over time, while weak evidence of an association was found for the trajectory of exposure to emotional violence.

The reduction in the prevalence of violence and harmful substance use at endline compared to baseline may be attributed to the impact of the COVID-19 pandemic. Sex work and its associated risks are known to expose FSWs to various stressors [19]. However, the measures implemented in Kenya to curb the COVID-19 pandemic, such as the closure of sex work hotspots, curfews, and social distancing from March to April 2020, affected the sex work industry, potentially reducing the prevalence of sex work-related stressors among FSWs in the study

**Table 4. Associations between the trajectories of violence and alcohol use problem with change in HCC at endline-multivariable regression.**

| Models | Variables | Adjusted GMR (95% CI) | P-value | Effect modification (P-value) | Did not attend counselling Adjusted GMR (95% CI) | Attended counselling Adjusted GMR (95% CI) |
|---|---|---|---|---|---|---|
| Model 1 | **Emotional Violence** | | 0.085 | | | |
| | Never | Reference | | 0.097 | Reference | Reference |
| | Baseline only | 0.69 (0.42 to 1.13) | | | 0.83 (0.42 to 1.62) | 0.45 (0.21 to 0.95) |
| | Endline | 0.58 (0.36 to 0.94) | | | 0.72 (0.33 to 1.56) | 0.37 (0.20 to 0.71) |
| Model 2 | **Physical Violence** | | 0.007 | 0.002 | | |
| | Never | Reference | | | Reference | Reference |
| | Baseline only | 0.59 (0.42 to 0.82) | | | 0.77 (0.48 to 1.22) | 0.39 (0.23 to 0.66) |
| | Endline | 0.63 (0.39 to 1.02) | | | 1.16 (0.51 to 2.65) | 0.36 (0.20 to 0.64) |
| Model 3 | **Physical and/ or sexual violence** | | 0.048 | 0.009 | | |
| | Never | Reference | | | Reference | Reference |
| | Baseline only | 0.71 (0.50 to 1.00) | | | 0.92 (0.58 to 1.46) | 0.46 (0.26 to 0.79) |
| | Endline | 0.63 (0.42 to 0.96) | | | 1.20 (0.61 to 2.35) | 0.37 (0.22 to 0.63) |
| Model 4 | **Alcohol use problem** | | 0.239 | 0.142 | | |
| | Never | Reference | | | Reference | Reference |
| | Baseline only | 0.77 (0.53 to 1.12) | | | 0.45 (0.26 to 0.79) | 1.23 (0.74 to 2.03) |
| | Endline | 1.11 (0.74 to 1.68) | | | 0.93 (0.51 to 1.70) | 1.31 (0.71 to 2.41) |

*__Models 1–4__ each adjusted for clinic, condom use at last sex, ACEs and number of children.

cohort. For example, as we have reported elsewhere from qualitative data, the closure of most sex work hotspots during the lockdown led to a decrease in clients, prompting some FSWs to pause or quit sex work and pursue other income-generating activities, such as selling masks, which may have reduced the overall prevalence of violent encounters with clients and law enforcement officers [48]. However, the qualitative findings also reported increased violence perpetrated by clients and law enforcement officers during the COVID-19 pandemic lockdown [48]. This reported increase in violence could reflect a rise in the severity and frequency of violence experienced by some FSWs, rather than an increase in the overall number of violent events experienced by FSWs in the study cohort. The reduction in violence observed in this study aligns with the quantitative findings from the larger Maisha Fiti study [49]. Additionally, the decline in the prevalence of alcohol and substance use reported in this study is plausible and may be linked to financial stress during the COVID-19 pandemic, as many FSWs could no longer afford alcohol and other substances [48].

The significant reduction in mean HCC levels at endline compared to baseline suggests changes in cortisol production among FSWs over time. Although the reasons for this decline are likely multifaceted, one possible explanation could involve long-term or severe stress exposure among participants, which may have led to HPA-axis dysfunction, resulting in lower HCC levels at endline, possibly following an initial increase [50]. We found a significantly lower HCC at endline for participants who experienced physical violence or physical and/or sexual violence at baseline only, compared to their unexposed counterparts, which could be plausible. Research shows that following a traumatic or stressful event, cortisol levels may initially increase and subsequently decrease below baseline levels [50]. Therefore, participants who experienced physical violence and physical and/or sexual violence at baseline but not at endline might show lower HCC levels at endline after an initial increase following baseline. The baseline cross-sectional findings from the same study cohort, reported elsewhere, revealed higher HCC levels among participants who had recently experienced physical and/or sexual violence (within the past six months) compared to their unexposed counterparts [34]. This suggests that the timing of chronic stress effects on cortisol levels is crucial, aligning with findings reported in a meta-analysis of chronic stress and

HPA axis in human [51]. However, since this study only measured cortisol levels at two-time points, further research with repeated measurements is needed to examine how HCC levels change over time with respect to the experience of violence. This would provide a more in-depth understanding of the impact of stress on HPA axis activation over time.

Additionally, the significantly lower HCC levels at endline among participants who experienced physical and/or sexual violence at endline (with or without baseline exposure) compared to those who weren't exposed also seems plausible. The lower HCC levels recorded could be due to severe/prolonged stress related to physical and/or sexual violence between the two-time points. Long-term chronic stress experiences eventually progress to HPA axis hypoactivity, as explained above, and the severity of violence has been linked to lower HCC levels [25,51,52]. Our findings highlight the potential adverse effects of physical violence or physical and/or sexual violence on the health of FSWs, as reduced cortisol levels can lead to poor health outcomes, including widespread inflammation, autoimmune diseases, and chronic pain [25]. Moreover, the lack of associations between the trajectories of exposure to any mental health problems and the consumption of harmful alcohol or other substances with a change in HCC levels at endline suggests that the observed change was not due to the experience of mental health problems or harmful substance use. However, these insignificant findings could be attributed to the limitations of this study as discussed below, including the sample size, reducing the power to detect any difference. Therefore, more research is needed [53].

Findings in this study showed the role of counselling in modifying the relationship between violence and change in HCC levels at endline. Amongst counselled participants, the stronger significant effect of physical violence or physical and/or sexual violence on reduced HCC at endline compared to those who did not attend counselling indicates that participants may have sought counselling due to the experience of physical and or sexual violence. Although we didn't find evidence of an association between attending counselling and change in endline HCC levels (Table 2), a recent review showed that HCC plays a vital role in evaluating psychological and neuropsychiatric interventions, with some studies recording decreased HCC after treatment [54]. Unfortunately, another limitation of our study was that we could not assess the effect of counselling services on HCC levels because participants were not randomised to counselling. However, based on the findings in this study in terms of counselling modifying the relationship between physical and or sexual violence and HCC, as well as the effectiveness of counselling interventions reported in other studies, there is a need for similar interventions to be embedded and maintained within existing services for FSWs.

## Strengths and limitations

A major strength of our study is the use of validated tools to assess our main exposures (violence, mental health, and alcohol and/or substance use), as well as the use of longitudinal data to assess the directionality of associations between our study exposures and change in HCC levels at endline. Using hair samples to measure HCC was another strength since it enabled us to measure long-term cortisol levels retrospectively. However, a key challenge in assessing HPA-axis dysfunction over time is the interpretation of the findings since cortisol can either increase or decrease depending on the timing or severity of the stressful event [50]. This could be one of the reasons why there is no established universal cut-off point for HCC measurement. Another limitation of this study includes the potential of under-reporting of sensitive issues such as condom use, mental health, and alcohol/substance use. Also, due to the retrospective nature of our key exposures, this study might be prone to recall bias. In addition, another key limitation was that only about 38% of HIV-ve women in this study had useable hair samples for both baseline and endline, which would have reduced the power of the study, biased the findings and limited generalisability. HIV status was one of the factors associated with loss of follow-up in the Maisha Fiti study, with lower follow-up among HIV-negative women [49]. Since this study focused on the HIV-negative women in the Maisha Fiti study, our findings may be prone to follow-up bias. However, the characteristics of the 425 HIV-negative participants with useable hair samples at baseline are similar to the 285 participants with follow-up useable hair samples. Moreso, our focus on women with useable 2 cm of hair meant the hair sample time frame (past 2.5 months based on African hair growth rate of 0.79 cm per month) [19] may not have matched the recent violence exposure

timeframes (past six months) or the mental health (past two weeks) time frames. This may have also caused bias in our estimates as the effect of violence on HCC, for example, may have been reduced or not fully captured if the incident of most violence occurred outside the hair sample time frame. Lastly, another major limitation of this study is our inability to capture the direct effect of COVID-19 infection on HCC levels, as there was a shortage of COVID-19 test kits in Kenya at the time of the study period.

## Conclusion

This study is the first to examine longitudinal changes in HCC levels among FSWs and to assess how the trajectory of exposure to violence, poor mental health, and harmful alcohol and substance use affect HCC over time. Findings show that the experience of physical violence and physical and/or sexual violence may lead to lower HCC levels over time. These suggest that the experience of physical violence and physical and/or sexual violence may lead to HPA axis dysfunction, which may serve as a potential pathway linking stressors to increased HIV risk and other health problems. However, further research with repeated measurement of HCC levels is needed to explore in-depth and, with a larger sample size, the associations between violence, HCC levels, and health status, especially HIV infection.

## Supporting information

**S1 Table. Comparison of the characteristics of study participants in this follow-up study (N = 285) with the study participants at baseline (425) and the HIV-negative participants in The Maisha Fiti study who were excluded at baseline (N = 321).**
(DOCX)

**S1 Checklist.**
(DOCX)

## Acknowledgments

We warmly thank all participants who contributed to this study, along with the SWARC (Sex Work Alliance Research Committee), who provided expertise on ensuring the research was beneficial to the sex work community. We also thank the Maisha Fiti Study Champions, who advocated for the Maisha Fiti study at the seven SWOP clinics, supported participants to participate at the study clinic, and answered questions in the field as they arose. The Maisha Fiti study champions were: Demtilla Gwala, Daisy Oside, Ruth Kamene, Agnes Watata, Agnes Atieno, Faith Njau, Elizabeth Njeri, Evelyn Orobi, and Ibrahim Lwingi. We also thank all Maisha Fiti research team members and our collaborators at Western University, London, Ontario, for conducting the cortisol ELISA analysis.

## Author contributions

**Conceptualization:** Mamtuti Panneh, Tara Beattie, Mitzy Gafos, John Bradley.

**Data curation:** Mamtuti Panneh, Polly Ngurukiri, Tanya Abramsky, James Pollock, Alicja Beksinska.

**Formal analysis:** Mamtuti Panneh.

**Funding acquisition:** Tara Beattie.

**Investigation:** Mamtuti Panneh, Tara Beattie, Qingming Ding, Rhoda Kabuti, Mary Kungu, Erastus Irungu, Janet Seeley, Helen A. Weiss, Abdelbaset A. Elzagallaai, Michael J. Rieder, Rupert Kaul, Joshua Kimani.

**Methodology:** Mamtuti Panneh, Tara Beattie, Qingming Ding, Rhoda Kabuti, Mary Kungu, Erastus Irungu, Janet Seeley, Helen A. Weiss, Rupert Kaul, Joshua Kimani.

**Project administration:** Rhoda Kabuti, Mary Kungu.

**Supervision:** Tara Beattie, Mitzy Gafos, John Bradley.

**Validation:** Rhoda Kabuti, Janet Seeley, Helen A. Weiss, Rupert Kaul, Joshua Kimani.

**Visualization:** Mamtuti Panneh.

**Writing – original draft:** Mamtuti Panneh.

**Writing – review & editing:** Mamtuti Panneh, Tara Beattie, Qingming Ding, Rhoda Kabuti, Polly Ngurukiri, Mary Kungu, Tanya Abramsky, James Pollock, Alicja Beksinska, Erastus Irungu, Janet Seeley, Helen A. Weiss, Abdelbaset A. Elzagallaai, Michael J. Rieder, Rupert Kaul, Joshua Kimani, Mitzy Gafos, John Bradley.

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
