## [Decision Letter · Decision Letter 0]

26 Mar 2025

PGPH-D-24-03042

Longitudinal assessment of changes in hair cortisol levels and associations with violence, poor mental health and harmful substance use among female sex workers in Nairobi, Kenya.

Dear Dr. Panneh,

Thank you for submitting your manuscript to PLOS Global Public Health. After careful consideration, we feel that it has merit but does not fully meet PLOS Global Public Health’s publication criteria as it currently stands. Therefore, we invite you to submit a revised version of the manuscript that addresses the points raised during the review process.

We look forward to receiving your revised manuscript.

Kind regards,

Joel Msafiri Francis, MD, MS, PhD

Academic Editor

Journal Requirements:

Additional Editor Comments (if provided):

Reviewers' comments:

Reviewer's Responses to Questions

**Comments to the Author**

1. Does this manuscript meet PLOS Global Public Health’s publication criteria?

Reviewer #1: Yes

Reviewer #2: Yes

2. Has the statistical analysis been performed appropriately and rigorously?

Reviewer #1: No

Reviewer #2: Yes

3. Have the authors made all data underlying the findings in their manuscript fully available (please refer to the Data Availability Statement at the start of the manuscript PDF file)?

Reviewer #1: No

Reviewer #2: Yes

4. Is the manuscript presented in an intelligible fashion and written in standard English?

Reviewer #1: Yes

Reviewer #2: Yes

Reviewer #1: This is a very nice manuscript exploring changes in hair cortisol concentrations in FSWs who are HIV negative and participating in a study in Kenya. The manuscript is interesting and well written, though I have some suggestions to improve it.

Abstract

1. Line 42 – how long after the baseline visit was the endline visit?

2. Line 51 – the term “trajectories” is vague – please describe more fully

3. There is no indication of the hypothesized direction of an effect nor of the results – the term “change” is used

Introduction

1. Nicely written overall.

2. Lines 133-146 do not hypothesize a main exposure variable, nor hypothesized directionality of changes. Did the authors expect increases or decreases? Or given the prior discussion about hyper- and hypo-cortisolism, maybe it is exploratory? This should be stated up front.

Methods

1. Please note the standardized scales within the text. I had to go to Table 1 to find it all.

2. Lines 256-260 – define alcohol and substance use “problems”

3. Give dates of the study, and time elapsed between the 2 visits

4. Given the large number of predictor variables of interest, would it be appropriate to consider using multiple comparisons for the p-values?

Results

1. Line 316 – Give breakdown of who did not have complete data from the parent study, and reasons why excluded, and examine if group differed.

2. Line 347 – Is it possible to present mean paired differences

3. Throughout – the term “never” threw me off. It is only for those 2 data points, so maybe make the name more informative, like “Neither timepoint”.

4. Similarly, the term “baseline only” in essence means a decrease in the variable (or absence of) from baseline to follow up, so I’d call that group “Decreased” or “Baseline: yes; Endline: no”.

5. “Endline (with/without baseline)” was more clear, but it would be clear to call them “Increased or sustained”.

6. I have a hard time interpreting the comparisons to the reference category, i.e. low risk at both timepoints. I would think the important comparison would be between sustained high risk and decreased risk. This is related to my prior comment about the lack of hypothesis, and the vague term trajectories. If these comparisons were made, the significant findings (which were all in comparison to the no risk at either time point), which is hard to interpret.

7. Line 385 – add the word “receiving” before counselling to remind readers that this was not a randomized state

8. Line 389 – in talking about the interaction, give the p-value. The stratified GMRs do not show the test for interaction.

Discussion

1. Lines 509-510. Move this to the results.

2. Lines 467-473 This sounds like a mediation claim, yet mediation was not analyzed, so the claim is unfounded.

3. Given the lack of a hypothesized direction of the HCC results, the discussion is somewhat muddled. Framing it either as exploratory, or giving a clear hypothesis would help the discussion flow.

4. I have a hard time interpreting the results given my comment #6 above.

Reviewer #2: A) OVERALL COMMENTS:

This paper addresses a highly relevant and timely topic with clear public health significance, particularly in the context of HIV epidemic control in sub-Saharan Africa, where this key population plays a central role. Indeed, from the body of literature available, it bridges the gap related to unavailability of longitudinal understanding of HCC levels in relation to the specified exposures of interest.

However, as a longitudinal or follow-up study published in Discover Mental Health (August 2024), it is standard practice in the research publication landscape to build on previously published work without repeating large sections from the original paper.

As noted under specific sections — including study design and sampling (Lines: 161–177), study variables (Lines: 242–253), and statistical analysis (from Line: 286 onward) — several sentences appear nearly identical to those in the original publication.

While it is commendable that the original study has been cited, I recommend a review of these sections to minimize repetition and improve conciseness. Streamlining this content will not only maintain academic integrity but also enhance the readability of the manuscript.

Otherwise, the manuscript is technically sound with the inferences being supported by data. The paper represents rigorous research both methodologically and from an ethical perspective. Also, the authors have made all data underlying the findings fully in line with PLOS GPH’s requirements.

B) MANUSCRIPT PRESENTATION & EDITORIAL COMMENTS:

In accordance with my review, the manuscript has been adequately presented in an intelligible fashion and written in standard and understandable English.

C) OTHER SPECIFIC COMMENTS:

Note: Below are the comments for specific sections and sub-sections (numbering is done for presentation purposes; it does not match what is in the original paper).

1) INTRODUCTION:

Given the established biological and epidemiological links suggesting that both COVID-19 infection and the broader psychosocial disruptions associated with the pandemic may contribute to chronic stress and dysregulation of the hypothalamic-pituitary-adrenal (HPA) axis, the author should consider enhancing the Introduction by including a brief discussion on the potential role of acute and long COVID-19 in elevating hair cortisol concentration (HCC). This addition is particularly important, as COVID-19 is later integrated into the conceptual framework. Although the study did not directly measure COVID-19 exposure or infection status, articulating this theoretical connection early in the manuscript would strengthen the rationale for its inclusion and provide a clearer conceptual foundation for the subsequent analysis (considering the fact that the original paper did not have COVID-19 included in the conceptual framework).

2) METHODS:

Please also

a) Study Design and Sampling

Line 176: Since both in the original paper (Reference No. 37) and this current one the analysis focused solely on the HIV-uninfected participants, this statement might make the reader think there were differences between the two. As the authors address the earlier highlighted recommendation, this specific statement should be looked at as well.

b) Data Collection: Behavioural-biological survey

• Line 188: The author refers includes in the list of time-invariant sociodemographic factors, age, marital status, and number of children. While in an infinitesimal period of time this might be true, in the actual sense they are liable to change.

• Line 192: I would like to point out an editorial comment that requires correction. The author mentions that "blood samples were used to test for syphilis using rapid plasma ‘regain’ assay". I believe this is a typo; it should read plasma ‘reagin’ assay. Please revisit and rectify accordingly.

3) DISCUSSION:

a) Strengths and Limitations

Line 493: While the author notes the use of validated tools to assess the main exposures, it is important to acknowledge a specific limitation of the WHO ASSIST tool - namely, its inability to accurately capture the quantity or dose of alcohol and substance use.

A related manuscript by the MAISHA team, published in Global Public Health (January 2024), reported that, in response to the COVID-19 pandemic, “To cope they [i.e., FSW] skipped meals, reduced alcohol use and smoking, started small businesses to supplement sex work or relocated to their rural home”.

In such contexts, the WHO ASSIST may detect a reduction in reduced frequency of alcohol and other substance use but not necessarily a reduction in quantity consumed per occasion. This distinction is critical, as reductions in frequency accompanied by continued high-dose consumption may still pose significant physiological risks, including elevated hair cortisol concentration (HCC). Therefore, alternative tools such as the WHO AUDIT (Alcohol Use Disorders Identification Test), which includes questions on both frequency and quantity of alcohol use, may have been more appropriate at least for measuring alcohol consumption.

Therefore, it is recommended that the author explicitly acknowledges this limitation in the manuscript to provide a more nuanced interpretation of the findings.

**Do you want your identity to be public for this peer review?** For information about this choice, including consent withdrawal, please see our Privacy Policy

Reviewer #1: **Yes: ** Judith Hahn, PhD

Reviewer #2: **Yes: ** Albert Komba

---

## [Editor Report · Decision Letter 1]

23 Dec 2025

Longitudinal assessment of changes in hair cortisol levels and associations with violence, poor mental health and harmful substance use among female sex workers in Nairobi, Kenya.

PGPH-D-24-03042R1

Dear Ms Panneh,

We are pleased to inform you that your manuscript 'Longitudinal assessment of changes in hair cortisol levels and associations with violence, poor mental health and harmful substance use among female sex workers in Nairobi, Kenya.' has been provisionally accepted for publication in PLOS Global Public Health.

Best regards,

Joel Msafiri Francis, MD, MS, PhD

Academic Editor